# Detecting Arboviruses Through Screening Asymptomatic Blood Donors in Rio de Janeiro/Brazil During a Dengue Outbreak

**DOI:** 10.3390/v17020224

**Published:** 2025-02-05

**Authors:** Marisa de Oliveira Ribeiro, Mônica Barcellos Arruda, Alexandre Rodrigues Calazans, Alexandre Vicente Frederico, Anielly Ferreira Brito, Beatriz Vasconcello de Souza Barreto, Élida Millena de Vasconcelos Brandão, Hamilton Athayde, Kátia Cristina Silva Nascimento, Luiz Paulo de Brito Oliveira Souza, Pedro Henrique Cardoso, Priscilla Lopes da Silva Guimarães, Vanessa Duarte da Costa, Carlos Alexandre da Costa Silva, Alexandra Martins Soares, Josiane Iole, Guilherme Louzada, Luiz Amorim Filho, Patrícia Alvarez

**Affiliations:** 1Laboratório de Kits Moleculares (LAMOL), Instituto de Tecnologia de Imunobiológicos (Bio-Manguinhos), Fundação Oswaldo Cruz, Fiocruz, Rio de Janeiro 21040-360, Brazil; mribeiro@bio.fiocruz.br (M.d.O.R.); monica.arruda@bio.fiocruz.br (M.B.A.); alexandre.calazans@fiocruz.br (A.R.C.); alexandre.vicente@bio.fiocruz.br (A.V.F.); anielly.brito@bio.fiocruz.br (A.F.B.); beatriz.barreto@bio.fiocruz.br (B.V.d.S.B.); elida.brandao@bio.fiocruz.br (É.M.d.V.B.); hamilton.barros@bio.fiocruz.br (H.A.); katia.nascimento@bio.fiocruz.br (K.C.S.N.); luiz.brito@bio.fiocruz.br (L.P.d.B.O.S.); pedro.cardoso@bio.fiocruz.br (P.H.C.); priscilla.guimaraes@bio.fiocruz.br (P.L.d.S.G.); palvarez@bio.fiocruz.br (P.A.); 2HEMORIO, Instituto Estadual de Hematologia, Rio de Janeiro 20211-030, Brazil; carlosalexandre113@gmail.com (C.A.d.C.S.); maralexandrasoares@gmail.com (A.M.S.); josiole@hotmail.com (J.I.); guilhermelouzada80@gmail.com (G.L.); luiz.amorim@hemorio.rj.gov.br (L.A.F.)

**Keywords:** arboviruses, blood donors, hemocenter

## Abstract

Arthropod-borne viruses (arboviruses) dengue (DENV), chikungunya (CHIK), and Zika (ZIKV) have been responsible for a high number of outbreaks worldwide. However, their screening in blood donors is not mandatory, and asymptomatic cases might act as an important cause of virus transmission via transfusion. A study was conducted to assess the presence of DENV (serotypes 1–4), ZIKV, and CHIKV in pooled samples (pool size: six) from asymptomatic blood donors. A total of 9463 plasma pools, corresponding to 56,778 blood donations from asymptomatic blood donors who attended donor sessions at HEMORIO and other blood centers in Rio de Janeiro and Espírito Santo, was submitted to automated nucleic-acid extraction and PCR amplification using ZC D-*Tipagem* molecular assay (Bio-Manguinhos). In general, a pool prevalence of 1% (95/9463) and a donor prevalence of 0.17% (95/56,778) were observed. January and February 2024 had a total of 62 positive pools out of 95 (65.3%). Targets DENV-1 and -2 had a higher prevalence in the studied months—early summer—with 24 and 28 positive pools, respectively. ZC D-*Tipagem* molecular assay was able to detect the best-known arboviruses circulating in asymptomatic blood donors; this study suggested that ZIKV, CHIK, and DENV are circulating in asymptomatic blood donors before blood donations and can be transmitted to blood transfusion recipients.

## 1. Introduction

Arthropod-borne viral (arboviral) infections are a group of viral infections spread to individuals by the bite of hematophagous insects such as mosquitoes, sandflies, and ticks [1,2]. In recent years, the occurrence of vector-borne diseases has accounted for more than 17% of all infectious diseases, culminating in more than 700,000 deaths annually [3]. The quantity of cases has increased in both endemic and non-endemic regions, especially given unorganized urbanization and climate changes [4,5].

Among the best-known arthropod-borne viruses (arboviruses), dengue, chikungunya, and Zika stand out. Dengue is one of the most prevalent vector-borne diseases worldwide, with more than 3.9 billion people in over 129 countries at risk of being infected by dengue virus (DENV). DENV is classified into four genetically distinct serotypes named DENV 1–4 [6]. In 2024, Brazil reported a higher number of dengue cases, with >6.0 million probable cases and more than six thousand deaths as of 31 December 2024; considering Rio de Janeiro, 302,552 probable cases were detected up through December [7]. Chikungunya virus (CHIKV), a member of the genus *Alphavirus*, is transmitted by *Aedes mosquitoes* (e.g., *A. albopictus*, *A. aegypti*). Outbreaks have been reported in South and Central America [8]. In 2023, 122 deaths were reported in Brazil, while 213 deaths were detected in 2024 [7]. Zika virus (ZIKV), first isolated in 1947 from a sentinel rhesus monkey [9], has been associated with the emergence of microcephalia in neonates (congenital Zika syndrome) and with Guillain–Barré syndrome [10].

There is little data in the literature on the prevalence of DENV, ZIKV, and CHIKV in blood donors. The occurrence of asymptomatic cases represents an important source of virus dissemination. Blood donor screening for circulating virus strains is an important approach for epidemiological surveillance. A previous study suggested a West Nile Virus transmission by transfusion and represented an incentive for the development of nucleic acid-based screening of blood donors [11]. Another study using nucleic-acid testing (NAT) in more than 400,000 blood donations identified five cases positive for ZIKV [12]. However, no post-transfusion infections were observed. Given this data, questions arise about the risk of these transmissions and the possible clinical consequences for receptors. The aim of this study was to assess the presence of DENV (serotypes 1–4), ZIKV, and CHIKV in pooled samples from asymptomatic blood donors from the states of Rio de Janeiro and Espírito Santo in Brazil using ZC D-*Tipagem* molecular assay (Bio-Manguinhos).

## 2. Materials and Methods

### 2.1. Sample Description

Convenience sampling was collected between December 2023 and May 2024 from asymptomatic blood donors who attended HEMORIO donor sessions, as well as donor sessions in 31 other blood establishments in the two states (Rio de Janeiro and Espírito Santo), both in Southeast Brazil. This research was performed in accordance with the Nuremberg Code involving rules with humans based on ethical principles. Ethical approval was not required in this study since the molecular evaluation was done based on Brazilian local or national regulations (Ordinance nº 158/2016 from the Brazilian Ministry of Health). A total of 9463 pools containing six samples each (56,778 samples) were analyzed using Laboratory of Molecular Kits (LAMOL) from Bio-Manguinhos in the HEMORIO blood center. Used at first for HEMORIO routine testing, these pools had already been tested in an initial screening using NAT PLUS HIV/HBV/HCV/Malaria Bio-Manguinhos (ANVISA, MH registration: 80142170056–RE No 863) [13], in which the results turned out to be negative for HIV, HBV, HCV, and malaria; therefore, a limited acid nucleic volume was sent for LAMOL and/or tested on the HEMORIO premises.

### 2.2. RNA Extraction

Plasma samples (1000 μL) and an internal control (IC) were submitted to nucleic-acid extraction using Chemagic™ Prime™ (Revvity, Waltham, MA, USA) from the Brazilian NAT PLUS platform, used for screening blood donors in the HEMORIO NAT laboratory.

### 2.3. Reverse Transcription and PCR Amplification

A reverse transcription associated with a real-time polymerase chain reaction (RT-qPCR) was done with a ZC D-*Tipagem* molecular assay for 32 reactions (Bio-Manguinhos). The assay was designed to differentiate DENV (serotypes 1–4), ZIKV, and CHIKV in a single reaction. A set of MGB-probes with primers aimed at a 3′ non-coding region (3′ NCR) of DENV (FAM—D1 and D3; ROX—D2 and D4), nonstructural protein 1 (NSP1) of CHIK (FAM), and nonstructural protein 1 (NS1) of ZIKV (VIC). The reaction was performed based on three modules combining targets and IC: CHIK/ZIKV/Ribonuclease P (RNAse P), D1/D2/IC, and D3/D4/IC. The IC used in DENV modules corresponds to a virus-like particle (VLP) protected by patent (PI0600715-5). Each pool of six samples was tested in three modules in different wells.

The RT-qPCR mixture contained 3.75 µL of Pathogen Reach Plus (PRP) Master mix, 1.25 µL of probe-associated primers, and 10 µL of viral nucleic acid extracted. The conditions for the RT-qPCR in the 7500 Real-Time PCR System (Applied Biosystems™, Waltham, MA, USA) were as follows: 50 °C for 15 min for reverse transcription followed by enzyme activation at 95 °C for 3 min, and then 40 cycles were conducted at 95 °C for 15 s and at 60 °C for 40 s. Ct values equal to or below 40 for DENV and CHIK and 38 for ZIKV represented positive results.

## 3. Results

In this study, a total of 9463 pools (56,778 samples) were tested by ZC D-*Tipagem* molecular assay; Table 1 shows the quantity of positive pools according to the respective month when samples were collected. In general, a pool prevalence of 1% (95/9463) was observed for positive pools, which corresponds to a 0.17% (95/56,778) RNAemia prevalence for blood donors. The Minimum Infection Rate (MIR) was 17 (95 positive pools/56,778 total processed samples × 1000).

The results suggested a higher arbovirus detection during January and February 2024 (during the early and warmest summer months), totaling 62 positive pools out of 95 (65.3%). Also, targets DENV-1 (24 pools) and -2 (28 pools) had a higher prevalence in those months. It is important to highlight the detection of ZIKV and CHIKV in three and six pools, respectively. Considering the months of January and February 2024, the dynamics of the targets’ detection throughout the days are indicated in Figure 1. It is observed that arbovirus detection occurred in half of the days of each month, especially for the DENV-1 and DENV-2 targets. ZIKV and CHIK were also detected in these months. There was one co-infected sample (DENV-1 + DENV-2).

## 4. Discussion

The prevalence of vector-borne diseases has grown considerably due to globalization, an increase in human travel, and continent-wide commerce. The number of arbovirus cases has increased in endemic regions, but new regions have also become prominent [5,14]. Arboviruses are considered a significant threat to blood safety, given their potential for transfusion transmission. ZIKV, CHIKV, and DENV are noteworthy for their dissemination capacity and global occurrence [15]. The emergence of these viruses represents a substantial risk to blood safety.

In the present study, we found an overall 1% arbovirus prevalence in the tested pool of six samples; the RNAemic donor prevalence was 0.17% (95 in 56,778 donors). The MIR was 17, i.e., it is assumed that 17 infected individuals existed in a positive pool and reflects the lower limit of true detection. This prevalence was significantly lower than the one seen in Rio de Janeiro’s blood donors during the last large dengue outbreak in Rio in 2012. The donor RNAemic prevalence in that 2012 epidemic was 0.54% (100% of DENV-4 cases) [16].

The current prevalence of dengue in viremic blood donors was also much higher than the one reported in a study from Northern Brazil: 36,133 blood donations (0.002%) [17]. Also, a comprehensive Brazilian study involving 994,370 donations from São Paulo, Belo Horizonte, Rio de Janeiro, and Recife found 1.2% of DENV RNAemic donation prevalence in Belo Horizonte in May 2019, but the incidence in Rio de Janeiro was only 0.0022%, i.e., more than 400 times lower than the 2024 figures [18]. In summary, a quantity of 95 positive samples from asymptomatic blood donors out of 156,778 tested shows that the risk of acquiring dengue through blood transfusion is not negligible, which indicates that donor screening could be justified as an important safety measure, at least during large outbreaks. A recent Brazilian study highlighted the necessity of increased investment in blood safety procedures to understand the actual incidence of transfusion-transmitted dengue virus (TT-DENV), given the death of four children in cardiovascular intensive care units. TT-DENV was considered the most likely source of infection, given the severe control measures adopted [19].

Our study also pointed out three positive cases for ZIKV; the study published by Custer et al. (2023) [18] did not identify any case of ZIKV in Rio de Janeiro’s blood donors in 2018 and 2019 samples. The prevalence (5.3/100,000), albeit low, can be a warning sign for Zika re-emergence in Rio de Janeiro; the epidemiological bulletin from July 2024, published by the Brazilian Ministry of Health [20], corroborates our findings. It reported two ZIKV cases in Rio de Janeiro’s general population during 2023 and 26 cases in the first part of 2024.

The prevalence of CHIKV viremic donors was 1:10,000; this is higher than the prevalence found by Custer et al. (2023) [18] in 2018–2019 (45:1,000,000) and lower than the incidence in Rio de Janeiro’s general population in 2024 (21.5:100,000). This discrepancy can be explained by the fact that CHIKV infections are rarely asymptomatic compared to DENV and ZIKV infections (around 10% for CHIKV versus 59.6% for DENV and 50% for ZIKV) [21,22,23,24]. When the rate of asymptomatic cases is low or very low, the efficacy of prospective blood donor deferral during the pre-donation interview is high, which can explain the discrepancy between blood donor prevalence and prevalence in the general population.

A recent meta-analysis reported and analyzed the prevalence of ZIKV, CHIKV, and DENV worldwide. Interestingly, the country with the highest prevalence estimate for ZIKV was Brazil (0.5%; 44/8578) [15]. Also, in April 2016, ZIKV RNAemia was detected at a prevalence of 638/100,000 donations (0.6%) in Hemominas and 639/100,000 donations (0.6%) in HEMORIO [18]. Several factors may have influenced the emergence of ZIKV in Brazil. The higher circulation of *Ae. aegypti* and *Ae. albopictus* vectors probably facilitated the ZIKV emergence [1]. The present study identified all four DENV serotypes circulating in Rio de Janeiro, although we just found one case of DENV-4, probably because of the remaining seropositivity serotype. A pool of six samples had an amplification of DENV-1 and DENV-2 simultaneously. These findings might be understood as such that two individuals were infected with one target separately, or a donor might have been co-infected with both targets. Interestingly, ZC D-*Tipagem* molecular assay (Bio-Manguinhos) was able to detect both viruses. It is important to highlight that a DENV epidemic was occurring in Brazil during the analyzed period since dengue has a seasonal profile with a possible increase of cases and risk for epidemics between October and May [7]. These data corroborated the present results since the prevalence of DENV was higher compared to other targets. Also, climate represents an essential influence on dengue temporal distribution [25]. Summertime in Brazil runs from December to March, but in 2024, high temperatures persisted even after summer. Therefore, relevant epidemiological data have been demonstrated with the present results. These findings raise attention to the safety of blood transfusion in hemocenters, especially in an epidemic country such as Brazil.

According to data from the Brazilian Ministry of Health, during 2024’s epidemiological weeks one and nine (January and February), more than 1.5 million individuals were considered DENV-probable cases in Brazil. Especially in Rio de Janeiro/Brazil, 130,176 individuals could possibly have a dengue diagnosis [7]. These data corroborate a higher arbovirus detection in the months of January and February 2024, since the number of suspected people with DENV increased considerably if a comparison with December 2023 (2023 epidemiological weeks 49–52) is made, during which fewer possible cases were reported (76,387). The number of probable cases decreased significantly in 2024’s epidemiological weeks 23–26, which corresponds to June (276,907) (Table 2); therefore, a lower virus circulation might have initiated in May 2024, as indicated by the present results in which only two pools had a DENV-2 detection in the stated month.

## 5. Conclusions

In summary, the ZC D-*Tipagem* molecular assay was able to detect the best-known arboviruses circulating in asymptomatic blood donors who attended the HEMORIO blood center in Rio de Janeiro. This study might have had limitations in terms of not describing the clinical impact of possible arbovirus infection after blood donation; however, the present research highlights the importance of testing for ZIKV, CHIK, and DENV before blood donations in Brazilian blood centers, given the fact that 1% of pools were contaminated by them. In conclusion, the ZC D-*Tipagem* molecular assay has significant potential in reducing transfusion-transmission arbovirus.

## Figures and Tables

**Figure 1 viruses-17-00224-f001:**
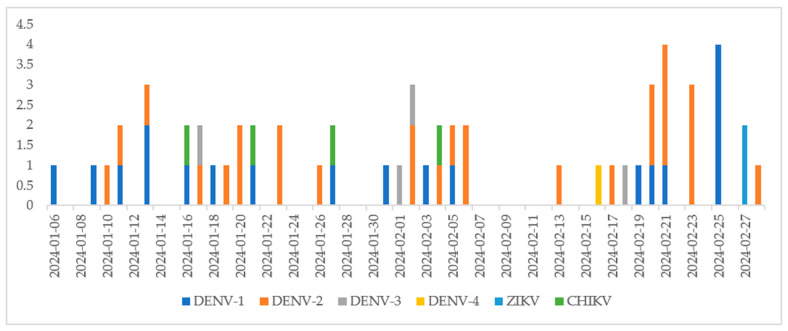
Distribution of detected targets throughout January and February 2024.

**Table 1 viruses-17-00224-t001:** Distribution of positive pools by month and their respective detected target.

				Targets (Positive Pool by Target)					
Month/Year	Number of Pools	Number of Samples	Positive Pools	ZIKV	CHIKV	D1	D2	D3	D4
December 2023	179	1074	1	-	-	-	1	-	-
January 2024	3291	19,746	26	-	3	12	10	1	-
February 2024	2788	16,728	36 *	2	1	12	18	3	1
March 2024	745	4470	15	1	1	9	3	1	-
April 2024	1633	9798	15	-	1	8	6	-	-
May 2024	827	4962	2	-	-	-	2	-	-
Total	9463	56,778	95 *	3	6	41	40	5	1

Legend: D1: DENV-1; D2: DENV-2; D3: DENV-3; D4: DENV-4. * One pool had an amplification of two targets simultaneously (D-1 and D-2).

**Table 2 viruses-17-00224-t002:** Probable dengue cases between December 2023 (epidemiological weeks 49–52) and June 2024 (epidemiological weeks 23–26).

Month/Year	Epidemiological Week	Number of Probable Cases
December 2023	49–52	76,387
January 2024	1–5	497,340
February 2024	6–9	1,116,098
March 2024	10–13	1,581,219
April 2024	14–18	1,885,120
May 2024	19–22	940,567
June 2024	23–26	276,907

## Data Availability

Data is contained within the article.

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
