# Peer review of "Detecting Arboviruses Through Screening Asymptomatic Blood Donors in Rio de Janeiro/Brazil During a Dengue Outbreak"

_viruses, 2025, doi:10.3390/v17020224_

Round 1
Reviewer 1 Report
Comments and Suggestions for Authors
Authors carried out a large-scale sero-epidemiological analysis for DENV, ZIKV, and CHIKV. The analysis was based on multiprex RT-qPCR with pooled samples. The data provides important knowledge about the presence of viremic bloods in healthy donors.
[Major comments]
It would be better to calculate minimum infection rate (MIR), adjusted infection rate (AIR) or maximum Likelihood Estimation (MLE). Since the authors discuss about the infection rate of those arboviral diseases, including MIR or other comparable data would strengthen the analysis.
[Minor comments]
In page 3 (2nd paragraph of the result section), "figure 2" should be revised to "figure 1".
In figure 1, auxiliary lines for 0.5x should be removed, as the data consists of natural numbers.
Reviewer 2 Report
Comments and Suggestions for Authors
The author tested 56,778 blood samples from 32 blood donation departments using a reagent of “ZC_D-Tipagem molecular assay (Bio-Manguinhos)” in a mini-pool of 6 samples during summer time in Brazil. A total of 95 pools were tested positive. The data indicates an un-neglectable risk of TT-arboviruses. This study is well-designed and the results presented are interesting, which are useful to further ensure blood safety.
Here are some comments:
1. In 2.3, the author mentioned “A reverse transcription associated with a real-time polymerase chain reaction (RT-qPCR) has been done with ZC D-Tipagem molecular assay for 32 reactions.”. what is the meaning of “32 reactions”? you mean minipools? Please clarify.
2. In 2.3, The author described the reagent probes used in this experiment were as follows: (FAM – D1 and D3; ROX – D2 and D4); CHIK (FAM); ZIKV (VIC). Different combinations of probes and internal controls were used to detect six targets, including D1/D2/D3/D4, ZIKV, and CHIKV. Are the three combinations of CHIK/ZIKV/Ribonuclease P (RNAse P), D1/D2/IC, and D3/D4/IC detected separately? If so, then each mini-pool would require three reaction wells. This point should be clarified in the manuscript, as mixing them in one reaction well would make it difficult for the fluorescent probes to differentiate between D1 and D3, etc.
3. In discussion,5th paragraph , the prevalence of CHIKV viremic donors was 1:10,000; this is higher, not lower than the prevalence found by Custer et al (2023) [18] in 2018-2019 (45:1000,000).
4. The last paragraph of discussion, the author described the prevalence of dengue in 2023 and 2024, maybe it is better to give the compared data in a form of table or bar-chart.
